# Anapole nanolasers for mode-locking and ultrafast pulse generation

Juan S. Totero Gongora[1], Andrey E. Miroshnichenko[2], Yuri S. Kivshar[2] & Andrea Fratalocchi[1]

Nanophotonics is a rapidly developing field of research with many suggestions for a design of nanoantennas, sensors and miniature metadevices. Despite many proposals for passive nanophotonic devices, the efficient coupling of light to nanoscale optical structures remains a major challenge. In this article, we propose a nanoscale laser based on a tightly confined anapole mode. By harnessing the non-radiating nature of the anapole state, we show how to engineer nanolasers based on InGaAs nanodisks as on-chip sources with unique optical properties. Leveraging on the near-field character of anapole modes, we demonstrate a spontaneously polarized nanolaser able to couple light into waveguide channels with four orders of magnitude intensity than classical nanolasers, as well as the generation of ultrafast (of 100 fs) pulses via spontaneous mode locking of several anapoles. Anapole nanolasers offer an attractive platform for monolithically integrated, silicon photonics sources for advanced and efficient nanoscale circuitry.

[1] PRIMALIGHT, Faculty of Electrical Engineering, Applied Mathematics and Computational Science, King Abdullah University of Science and Technology, Thuwal 23955-6900, Saudi Arabia. [2] Nonlinear Physics Centre, Research School for Physics and Engineering, Australian National University, Canberra Australian Capital Territory 0200, Australia. Correspondence and requests for materials should be addressed to Y.S.K. (email: yuri.kivshar@anu.edu.au) or to A.F. (email: andrea.fratalocchi@kaust.edu.sa).

In recent years, significant efforts have been devoted to realize nanoscale sources of coherent light aiming at filling the gap between photonics and electronics for both classical and quantum applications[1–17]. Fundamental advances have been recently reported in semiconductor–dielectric heterojunctions, where nanolasers from direct bandgap materials have been demonstrated[18–20]. Further progress with these materials requires to address many fundamental challenges, originating when the miniaturization of optical circuitry is pushed to the nanoscale. A first problem comes from the diffraction limit of light that introduces significant radiative losses that are detrimental and severely limit the performance of nanoscale devices[21]. A second issue is represented by the efficient coupling of light from nanophotonics sources, whose emission is strongly nondirectional[22]. All these problems originate from the fact that conventional optical sources couple energy to classical radiating states that are all diffraction limited and difficult to control at the nanoscale.

An intriguing idea is to introduce a nanolaser design concept based on states of matter with unconventional radiation properties that can overcome the aforementioned limitations. States with non-conventional emission of energy have been recently demonstrated in specific types of all-dielectric nanoparticles, realized in both silicon and germanium semiconductors[23–25]. Under specific conditions, the superposition of internal modes of these nanostructures can generate radiation-less states, called anapoles, that are scattering-free and invisible to the propagation of the electromagnetic fields[26,27].

A fundamental question is whether it is possible to employ a non-radiating state of matter, such as an anapole, to engineer a coherent light source based on stimulated light amplification. The possibility to achieve nonlinear amplification of an anapole state is counterintuitive, due to the lack of any optical feedback from a scattering-free state that is totally invisible to a far-field observer. However, if such interaction could be successfully triggered, we would have at disposal a new type of source that is not limited by diffraction or any of the aforementioned problems.

By using rigorous first-principle simulations on the quantum electrodynamics of photons, here we show that the light–matter interaction in a nanolaser emitting at the anapole frequency gives rise to a surprisingly stable steady state, where light energy is strongly collected within the anapole and evanescently transmitted in a subwalength area outside the nanostructure. By leveraging on tunnelling effects and mutual synchronization, we illustrate that such 'anapole nanolaser' opens a large manifold of applications, ranging from efficient and polarization controlled energy coupling, to ultrafast pulse generation without the need of external design elements. Anapole nanolasers hold the promise to be an ideal energy source for nanoscale optics in silicon-compatible platforms.

## Results

**Design of an anapole-based nanolaser.** To illustrate the concept of anapole-based nanolaser (see Fig. 1), we consider an all-dielectric resonator composed of a semiconductor nanodisk that is optically pumped in the same scheme of a classical laser[28–31], with energy decaying in one -or more- indirect transitions to a final energy state (see Fig. 1b). Unlike a conventional laser, however, in our design the stimulated emission (see Fig. 1b red arrow) amplifies an anapole state, which can be identified by a dip in the scattering cross section of the structure (Fig 1c, blue line). The anapole mode is generated by the superposition of electric and toroidal dipoles[24], which produces an electromagnetic mode that is confined in the interior of the nanodisk and does not radiate in the far-field (see Fig. 1d). In the scheme illustrated in Fig. 1, the energy down-converted by stimulated emission at the anapole frequency excites electric and toroidal dipoles that interfere destructively in the far-field. Contrary to lasers based on dark modes, this scheme is not based on resonant states, nor does it require non-radiative energy transitions in order to allow the generation of radiation-less states in purely dielectric materials[28,29]. In contrast to bound states in the continuum (BIC) lasers recently proposed[30,31], the anapole laser does not require extended structures and the radiation-less state is excited with no additional radiating components.

We design an integrated anapole source starting from standard $In_xGa_{1-x}As$ semiconductor that is obtained by combining InAs and GaAs and is a direct bandgap semiconductor that is silicon compatible for monolithic integration[19]. Following the electrical or optical excitation of carriers, the materials sustain stimulated emission of radiation. Both the optical properties and the emission frequency can be tuned by varying on the molar concentration $x$ of InAs in the compound[32]. Figure 2a illustrates the bandgap energy of $In_xGa_{1-x}As$ as a function of the Indium molar fraction $x$. As shown in the figure, the emission of the material can be tuned in a large bandwidth from 873 nm in the near infrared to 3.6 μm in the infrared. In our analysis, we consider a molar fraction $x = 0.15$, corresponding to an emission wavelength $\lambda_0 = 948$ nm. The value $x = 0.15$ is a standard concentration of InAs that allows monolithic integration directly on silicon, as reported in ref. 18. In addition to that, $In_{0.15}Ga_{0.85}As$ is characterized by low losses in the proximity of the emission frequency $\lambda_0$, as illustrated in Supplementary Fig. 1.

Figure 2b illustrates the scattering cross-section $C_{sca}$[33] of a nanodisk of $In_{0.15}Ga_{0.85}As$ with height 100 nm and different values of diameter $d$. The figure shows a region where the scattering of the nanodisk is suppressed in all directions (see Fig. 2b dashed line). Such special region, which extends in the visible and in the near infrared for $In_{0.15}Ga_{0.85}As$ nanodisks, is sustained by the presence of non-radiating anapole states that do not possess far-field emission. This is further confirmed by the multipole decomposition of the electromagnetic fields as a function of the incident wavelength for a nanodisk with diameter $d = 440$ nm and height $h = 100$ nm (Supplementary Figs 2–5).

Figure 2d presents the scattering cross-section spectrum for a nanodisk with diameter $d = 440$ nm and height $h = 100$ nm, whose anapole wavelength coincides with the emission wavelength of the $In_{0.15}Ga_{0.85}As$ semiconductor at 948 nm (red arrow). This configuration constitutes our design geometry as illustrated in Fig. 1c. To maximize the energy trapped inside the nanodisk for the pump beam, the pump wavelength can be placed on a scattering maximum of the nanodisk (see Fig. 2c, green arrow). In the proximity of the anapole frequency, the scattering of the structure is strongly suppressed within a bandwidth of $\Delta\lambda \approx 50$ nm. This is larger than the amplification bandwidth of the $In_{0.15}Ga_{0.85}As$ semiconductor that is ∼20 nm (see Fig. 2d, red solid area) as experimentally measured in refs 18,34. The anapole state is stable against geometrical changes in the structures, as we verified by calculating the scattering cross-section $C_{sca}$ of the structure for varying nanodisk diameter $d$ and height $h$ (see Supplementary Fig. 6). The formation of anapole states dominate at $\lambda_0 = 950$ nm in a wide range of geometrical parameters corresponding to 85 nm $\leq h \leq$ 105 nm and 435 nm $\leq d \leq$ 475 nm that are well within the precision of optical nano fabrication technologies such as electron beam lithography.

In the case of lattice mismatch between the anapole nanodisk and the substrate, we do not expect in general any perturbation in the optical properties of the system, as long as the effect is limited to few lattice sites. However, to reduce lattice-mismatch non-idealities, the InGaAs layer can be growth by using III–V

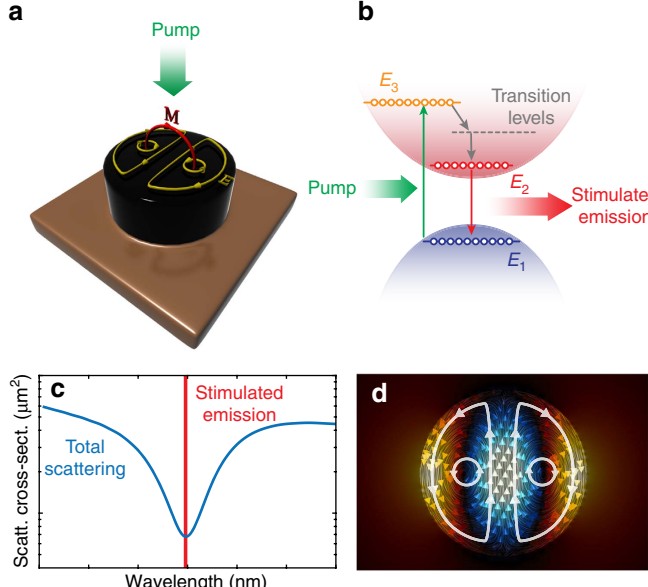

**Figure 1 | Concept of anapole nanolaser.** (**a**) Three-dimensional representation of an anapole nanolaser composed by a direct bandgap semiconductor (black nanodisk) that is optically pumped from the top (green arrow). (**b**) Energy diagram of light amplification via stimulated emission in a direct bandgap semiconductor. (**c**) Amplification scheme in frequency domain of the anapole nanolaser, where the stimulated emission (solid red line) coincides with the scattering dip (solid blue line) associated to an anapole state. (**d**) Spatial distribution of the electric field inside the nanodisk at the anapole frequency. The formation of circulating patterns (white arrows) suppresses far-field radiation and generates a scattering-free anapole state whose energy is confined inside the nanostructure.

semiconductors on a lattice-matched substrate, and then transferred on a low-refractive index substrate that is compatible with Si technology. At the proposed $x = 0.15$ concentration of Indium, the InGaAs is lattice matched to both InP and GaAs that could be used as ultrathin epitaxial growth substrates. By introducing a sacrificial layer (for example, AlAs), the InGaAs/ GaAs or InGaAs/InP substrate is easily transferred to a low-refractive substrate. Such technique is widely used in the fabrication of III–V semiconductor nanolasers on silicon platforms, as discussed in refs 35,36.

Our choice of InGaAs for the anapole laser also opens the possibility to design electrically pumped systems. Although this topic clearly falls beyond the scope of this article, some points can be highlighted. To realize an electrically pumped anapole laser, the main point is to design a semiconductor heterostructure that ensures three-dimensional (3D) confinement of the anapole mode. A possible solution is to consider a core–shell geometry. As discussed in ref. 26, anapole states can be excited in core–shell structures by acting on the multilayer thicknesses. By employing this strategy, an electrically pumped anapole laser can be composed of InGaAs/GaAs or InGaAs/InP. This structure, after adding top and bottom electrodes for electric carriers injection, can provide an initial setup for an electrically pumped anapole laser, where the anapole state is generated in the intrinsic layer InGaAs. Another important point is the optical footprint of the electrical circuitry. To this extent, dielectric conductors such as indium tin oxide are preferred to metal contacts. The use of dielectric contacts helps in minimizing optical losses. An intriguing possibility is the introduction of graphene-based contact layers or nanowires, such as the ones proposed in ref. 37, that is a promising design for a room-temperature electrically pumped anapole laser.

We observe that the anapole state is not a resonance of the system, therefore it is not possible to associate intuitive quantities such as quality factors which are employed in the description of resonant laser cavities. Resonant states are in fact observed in the spectral response as Lorentzian energy peaks with full width half maximum equal to $\omega_0/Q$, with $\omega_0$ being the resonant frequency. An anapole state does not possess a Lorentzian shape and is conversely observed in points of the spectrum where the scattering cross-section vanishes.

**Anapole amplification by stimulated emission of radiation.** We investigated the process of amplification of anapoles through stimulated emission by the Maxwell–Bloch finite-difference time-domain (MB-FDTD) (see Methods). This technique has proven to furnish extremely realistic results that have been verified against many experiments[38–41]. In our simulations, we modelled the exact dispersion curve of the $In_{0.15}Ga_{0.85}As$ (see Supplementary Fig. 1), and we assumed a Lorentzian-shaped gain profile with a dephasing time of $\tau = 40\,ps$, whose value has been experimentally measured in refs 18,34.

Figure 3 summarizes our 3D results for a pumping beam linearly polarized along the $x$ axis and perpendicular to the nanodisk axis. In these simulations we considered a nanodisk of $In_{0.15}Ga_{0.85}As$ with diameter 440 nm and height 100 nm, exposed to different pumping rates that are measured through the density of carriers $\rho_0$. Figure 3a,b report the intensity and the linewidth of the electromagnetic field of the anapole state amplified inside the nanodisk. The behaviour of the anapole amplitude versus pumping rate (see Fig. 3a,b) is that of a standard laser, with a linear relationship between pumping rate and amplified intensity (see Fig. 3a). As in plasmonic and dielectric nanoparticle lasers[18,42], the amplification of anapole states directly follows from stimulated amplification as a balance between field enhancement in the gain material and the system losses, including absorption and near-field energy leakage. The anapole laser also shows an equivalent Schawlow–Townes linewidth (see Fig. 3b) that quickly reaches a stable value of $\sim 2\,nm$. This value is in accordance with experiments on lasing emission from InGaAs nanostructures[18]. The carrier threshold density to lasing, observed in Fig. 3a, depends on the pumping rate $\rho_0$ that is proportional to the population inversion density, or equivalently, to the input pumping rate of the system. By increasing the latter, we increase $\rho_0$ and control the laser threshold condition.

A characteristic time evolution of the $E_x$ field is displayed in Fig. 3a (inset) that shows the reaching of a stable stationary emission state after an initial transient. The corresponding intensity spectrum, reported in Fig. 3b (inset), further confirms that the emission corresponds to the amplification of the anapole state at $\lambda_0 = 948\,nm$.

Figure 3c–e illustrates the spatial distribution of the electromagnetic energy of the system at steady state. Here we observe a radically different scenario if compared with a classical nanolaser. The emitted field is in fact totally confined within the nanodisk resonator, with electromagnetic energy decaying exponentially outside the nanodisk region. The lack of far-field emission provides a characteristic signature of the anapole state that is being amplified through stimulate emission, and is the hallmark of a new type of near-field nanolaser that does not radiate in the far field. Inside the nanodisk, we observe that the electromagnetic energy distribution preserves the dipolar shape of the linear anapole state (see Supplementary Fig. 7) that is perpendicular to the polarization of the pumping field $E_x$. We evaluated the correlation between the energy distribution of the anapole state amplified in Fig. 3c–e and that of an ideal anapole in a lossless InGaAs structure, obtaining a correlation amplitude of 96.1%.

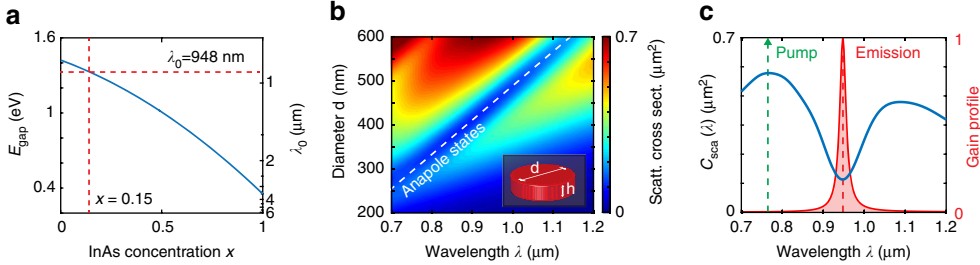

**Figure 2 | Design of anapole nanolaser.** (**a**) Tunability of the direct bandgap of $In_xGa_{(1-x)}As$ as a function of the molar ratio $x$ of InAs. (**b**) Pseudocolour plot of the scattering cross-section $C_{sca}$ of an $In_{0.15}Ga_{0.85}As$ nanodisk for a varying disk diameter $d$ and of the incident wavelength. The incident field is linearly polarized along the $E_x$ direction. The white dashed line highlights the region where the scattering is suppressed by the presence of anapole states. (**c**) Scattering cross-section for an $In_{0.15}Ga_{0.85}As$ nanodisk of diameter $d = 440$ nm and height $h = 100$ nm. For these choice of parameters, the anapole wavelength coincides with the semiconductor emission wavelength $\lambda_0$ (red arrow). The red shaded area shows the Lorentzian gain profile of the $In_{0.15}Ga_{0.85}As$ semiconductor that is entirely contained in the scattering suppression region.

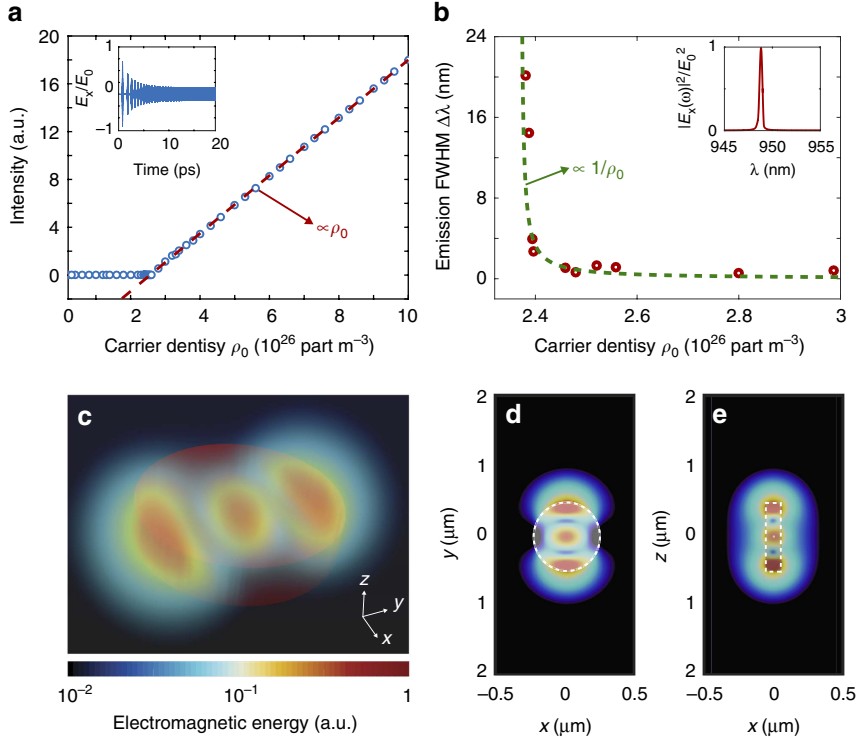

**Figure 3 | Near-field lasing action of anapole states.** (**a**) Input/output diagram of anapole amplitude versus carrier density $\rho_0$ with (inset) time trace (**a**) of the field component $E_x$ for a pumping rate of $\rho_0 = 3 \times 10^{26}$ part m$^{-3}$. (**b**) Corresponding linewidth behaviour and (inset) power spectral density measured inside the nanodisk. (**c–e**) Three-dimensional volume mapping (**c**) of the electromagnetic energy in the steady-state for $\rho_0 = 3 \times 10^{26}$ part m$^{-3}$. (**d,e**) show top (**d**) and lateral (**e**) sections of (**c**).

This shows that the anapole laser amplifies through stimulated emission an almost ideal radiation-less anapole state. Supplementary Note 1 and Supplementary Figs. 2–5 provide an in-depth analysis of the role of losses in the system.

**Spontaneously polarized nanolasers and photonics couplers.** The possibility to exploit near-field emission in conjunction with the dipolar nature of the anapole mode opens new pathways to develop spontaneously polarized nanolasers and nanophotonic routers. Polarization control, which is crucial in many linear and nonlinear applications[43–46], is quite a challenging task for nanoscale sources and it is in general addressed with the introduction of metals and/or specific geometries[47–49]. By employing anapole sources we can introduce a new design concept that can be easily integrated into two-dimensional

circuits. Figure 4 illustrates our idea, where we consider a nanodisk of $In_{0.15}Ga_{0.85}As$ with 440 nm diameter and 100 nm height placed in proximity of silicon nanowires of width 150 nm and thickness 100 nm. The nanowires are distributed in cross configuration, with the anapole nanolaser placed at the intersection point of the circuit. When the incident beam impinges on the anapole with linear polarization (see Fig. 4a), it excites the corresponding anapole state with localized dipolar field aligned perpendicularly to the pump polarization. This depends on the fact that the energy distribution of the anapole, resulting from the near-field interference between electric and toroidal dipoles, is strongly asymmetric in one spatial direction orthogonal to the input polarization (please see Supplementary Fig. 7). When the anapole is nonlinearly amplified by stimulated emission, the input pump polarization selectively excites the corresponding symmetry direction, leading to different energy

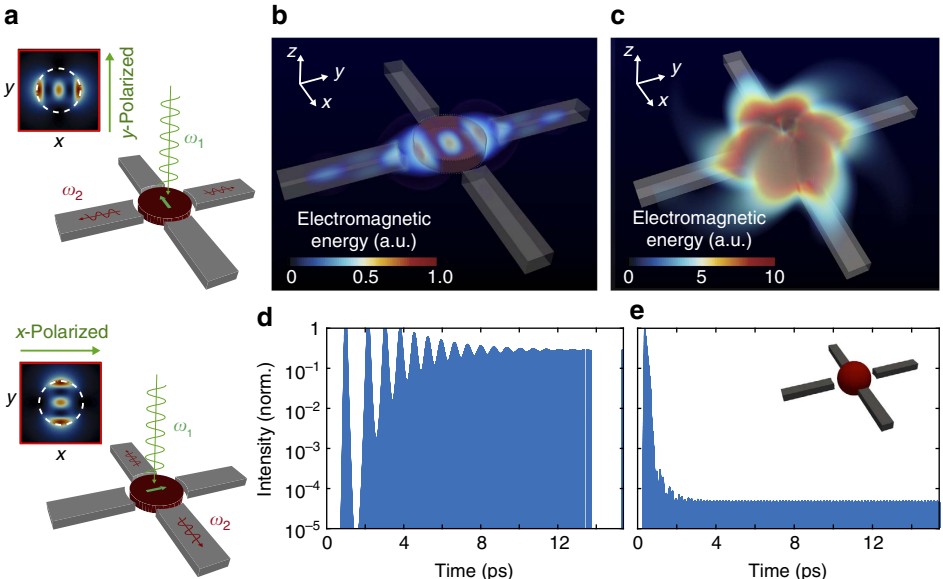

**Figure 4 | Polarization controlled and optical router anapole nanolaser.** (**a**) Working principle of the device based on directional coupling of anapole modes. Light pumping the anapole nanolaser placed in the middle of the circuit amplifies the corresponding anapole mode that tunnels energy in neighbouring channels according to the polarization of the input pump. (**b**) Electromagnetic energy distribution in the nanowire channel at steady state in the case of an $E_x$ polarized pumping beam. (**c**) Same structure as in **a**,**b**, but with an $In_{0.15}Ga_{0.85}As$ spherical nanoparticle replacing the anapole nanolaser (see (**e**), inset). The nanoparticle has the same radius as the nanodisk and is characterized by a strong resonance at the emission wavelength $\lambda_0$. The emission of the nanoparticle is more than one order of magnitude larger than the anapole, as observed by comparing (**b**,**c**). However, it does not selectively couple to any channel due to the strongly radiating character of the emission intensity. (**d**,**e**) Normalized electromagnetic intensity evolution inside the nanowire channels. The intensity is normalized to the intensity of the electromagnetic field inside the nanodisk. The intensity coupled in each nanowire channel from the nanospheres (**e**) is four order of magnitude lower than for the anapole nanolaser (**d**).

distributions in the final anapole state (see Fig. 4a). The electromagnetic field distribution of the anapole couples only into the channels that overlap with the anapole mode, routing light according to the polarization of the input field. Figure 4b reports a volumetric mapping of the spatial distribution of the electromagnetic energy at steady state and illustrate this dynamics. The near-field characteristics of the anapole allows electromagnetic energy to be efficiently coupled inside nanowire channels even of such a small size. To evaluate this functionality more quantitatively, we considered the same scenario with a standard nanolaser, characterized by a classical spherical nanoparticle with diameter $d = 448$ nm and resonance at 948 nm (Fig. 4e, inset). Figure 4c illustrates the outcome. Due to the strongly radiating nature of the components amplified in the nanoparticle, light scatters almost everywhere without coupling in the nanowire channels. By measuring at the light energy in the nanowire channels, we observe that with an anapole nanolaser we have four orders of magnitude more energy inside the channel with respect to the standard nanoparticle laser (see Fig. 4d,e). Such remarkable result depends on the fact that an anapole nanolaser enables by design near-field energy coupling, a mechanism that is extremely efficient in transferring energy in neighbouring structures. In Supplementary Fig. 8 we considered in detail the comparison with another integrated lasing structure, namely a nanodisk with the same size of the anapole laser but now set to emit at 1,125 nm. At this wavelength, the structure shows a strongly dipolar energy emission that well couples with the fundamental mode of the nanowire channels. In this case the energy couples inside all channels. The energy transferred in the waveguides is 15 times lower that the one observed with the anapole, showing the superior performances of the anapole laser.

The tunability of the formation of an anapole state in the $In_{0.15}Ga_{0.85}As$ material (see Fig. 2) and the compact geometry of the anapole near-field nanolaser allows a large design flexibility, where multiple anapoles can be optically pumped within hundreds of nm, feeding light energy into complex nanoscale circuits. In this scenario, even the use of a scanning near-field optical microscope nanotip or a plasmonic nanoantenna is very challenging, as the macroscopic size of the coupling structure does not easily allow the excitation of selective channels nearby in space.

**Anapole mode locking and on-chip ultrafast pulse generation.** The evanescent emission of energy from anapole nanolasers opens to new concepts for ultrafast pulse generation that results from a mechanism of mode locking that is based on the near-field synchronization of different anapoles. In photonics, synchronization phenomena that are usually exploited for the generation of optical pulses require expensive setup with specific design elements, such as $Q$-switching components, saturable absorbers and optical modulators, that cannot be miniaturized to the nanoscale.

Recent experiments reported the observation of mutual synchronization in space of laser sources emitting at the same frequency[50–52] that can reconfigure the spatial distribution of the emitted energy in the far field. These results open the question of whether it is possible to exploit the mutual synchronization in time of different anapole lasers, designing a new process of mode locking that generates pulses without the need of external design elements than the laser itself.

Figure 5a shows the design principle of our idea, where we consider a linear array of anapole nanolasers with slightly different radii. Due to the radius variation, each anapole possesses a different emission frequency $\omega_n = \omega_0 + n \cdot \Delta\omega$ that lies in the 20 nm amplification bandwidth of the $In_{0.15}Ga_{0.85}As$ semiconductor (see Fig. 2b). The collective emission from the anapole

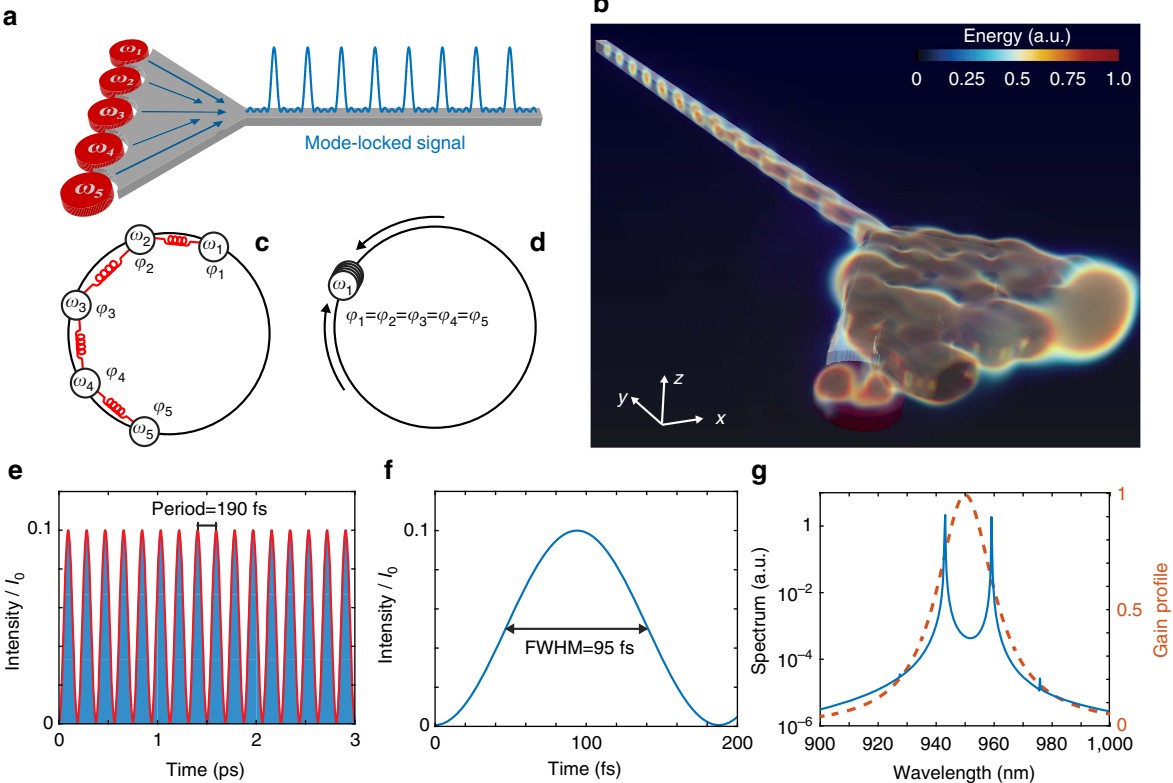

**Figure 5 | On-chip ultrafast pulse generation with anapole nanolasers.** (**a–c**) Integrated mode-locking device structure composed by an chain of anapole nanolasers with increasing emission frequencies $\omega_1, \omega_2, ..., \omega_n$ within the amplification bandwidth of the In$_{0.15}$Ga$_{0.85}$As semiconductor. The structure is coupled via tapered junction into a nanowire channel. (**b,c**) Provide a graphical illustration of the mechanism of pulse generation in this structure. Each anapole acts as a nonlinear oscillator characterized by a different emission frequency $\omega_n$ and phase $\varphi_n$. Due to nearest-neighbour interactions, anapoles tend to mutually lock their phases to the same value, generating a stable train of pulses in the channel waveguide. (**d**) The 3D volume mapping of the electromagnetic energy in the structure at steady state. (**e,f**) Intensity evolution inside the waveguide channel (blue line). The intensity is normalized to the intensity of the electromagnetic field inside the nanodisk. The system forms equispaced pulses with duration of 95 fs and period of 190 fs. The solid red line in **e** shows the theoretical prediction from the standard theory of mode locking. (**g**) Emission spectrum (solid blue line) superimposed to the semiconductor gain profile (dashed orange line) exhibiting the locking of two electromagnetic frequencies located on the edges of the semiconductor amplification band.

chain is collected by a tapered structure into a single-mode nanowire channel of width 150 nm and height 100 nm.

The anapole chain, in this scheme, behaves as a system of weakly coupled, nonlinear oscillators. Nonlinearity is here provided by the presence of the amplifying optical semiconductor. Weak coupling, conversely, arises from the spatial overlapping of the anapole emitted fields (see Fig. 3). Each anapole behaves initially as an independent oscillator at the frequency $\omega_n$ and phase $\varphi_n$, as illustrated in Fig. 5b. During the time evolution of the system, the frequencies $\omega_n$ and phases $\varphi_n$ interact are allowed to interact via nearest-neighbour coupling. To understand the dynamical outcomes of these interactions, we developed a simplified model of light–matter interaction by generalizing the theory of coupled laser systems[53] to the case of multiple anapole lasers emitting at different frequencies (see Supplementary Note 2 for a full derivation and for a detailed analysis of the model):

$$\begin{cases} \frac{dE_n}{dt} = (1 + i\alpha)N_n E_n + \sum_m C_{nm} E_m, \\ \frac{dN_n}{dt} = \mu\left[P - N_n - (1 + 2N_n)|E_n|^2\right], \end{cases} \quad (1)$$

where $E_n = A_n e^{i\phi_n}(t)$ is the complex electric field amplitude of the the $n$th anapole in the chain, $t = t/\tau_p$ a dimensionless time measured in units of the photon carrier lifetime $\tau_p$, $N_n$ the population inversion, $\alpha$ the linewidth enhancement factor, $\mu = \tau_p/\tau_s$

the ratio of photons $\tau_p$ to carrier $\tau_s$ lifetime, $P$ the pumping rate and $C_{nm} = c_{mn}e^{i\psi_{nm}}$ coupling coefficients, arising from spatial overlaps between weakly overlapping nearest-neighbour anapoles.

For $\alpha = 0$, equation (1) reduce to the well-known Kuramoto model, a universal model of synchronization dynamics that has been largely studied in the past[54]. At steady state, for increasing pumping rate $P$ and nonzero coupling between the anapoles, both amplitudes and phases tend to lock to a common value $A_n(t) \rightarrow A_0$, $\varphi_n(t) \rightarrow \varphi_0$. Supplementary Figure 9a,b shows an example of this dynamics. For a realistic anapole laser with nonzero $\alpha$, the frequency of oscillation of each electric field becomes influenced by the carrier population $N_n$. On a timescale smaller than $1/\mu$, no effect is practically observed and the anapole network behaves as a classical Kuramoto model. Conversely, for times larger than $1/\mu$ the synchronization conditions between the anapole oscillators change, with some anapoles getting detuned from the others. Supplementary Figure 9c,d illustrates an example, obtained for $\alpha = 1$ in an ensemble of 10 anapoles.

Synchronization events in the anapole chain open the interesting possibility to control the time dynamics of the laser at the ultrafast scale. When a synchronization event takes place, in fact, the total emission intensity $I(t)$ acquires the temporal profile of a train of pulses, expressed by the superposition of the

harmonics that becomes synchronized:

$$I(t) \propto \left| \sum_n E_n \right|^2 = \left| A_0 \sum_n e^{i\omega_0 t + n\delta\omega t} \right|^2 = \left| A_0 \frac{\sin\left(\frac{N\delta\omega t}{2}\right)}{\sin\left(\frac{\delta\omega t}{2}\right)} \right|^2. \quad (2)$$

The period of the train is $T = 2\pi/\delta\omega$ and the individual pulse width is $\Delta = 2\pi/N\delta\omega$. These quantities depend on the number $N$ and frequency spacing $\delta\omega$ of the anapoles that become synchronized. Following equation (1), the synchronization process is controlled by the pumping rate $P$ and coupling coefficients $C_{mn}$, as shown in Supplementary Fig. 9.

To validate this theoretical picture for a realistic anapole laser, we performed first-principle simulations on the anapole chain. Figure 5d–g shows the lasing behaviour of the anapole array. Figure 5d shows a snapshot of a 3D volume mapping of the energy distribution of the system, showing the near-field coupling of energy into the nanowire optical channel and the high degree of confinement ensured by the evanescent coupling of the anapoles radiation into the tapered structure. In the steady state, the anapoles mutually synchronize and the structure naturally generates train of pulses, as shown in Fig. 5e where we report the emitted intensity at the end of a 5 µm waveguide. Fig. 5f provides a zoomed view of a single pulse, characterized by an ultrashort duration $\Delta t = 95$ fs. The spectrum of the electric field (see Fig. 5g) well illustrates the locking of the various emitted frequencies around the amplification band edge of the $In_{0.15}Ga_{0.85}As$ semiconductor (see Fig. 5g dashed line). By extracting the mode spacing $\Delta\omega = 4 \times 10^{12}$ rad s$^{-1}$ and their spectral amplitudes, it is possible to compare the emitted intensity predicted by equation (2) that shows perfect agreement with the results from first-principle calculations (see Fig. 5e). Remarkably, the mode-locking mechanism is a stable and repeatable feature of the system, as we verified by performing a statistical campaign of 50 distinct simulations. Due to the stochastic properties of quantum noise, each simulation corresponds to a different set of initial conditions for the phases of the nanolasers along the chain.

The dynamical model described by equation (1) of the anapole chain suggests that the number of anapole that becomes synchronized can be changed by acting on the coupling matrix elements $C_{mn}$. This observation is stimulated by the similitude of equation (1) and the dynamics of oscillatory neural networks[55]. Simple calculations on equation (1) show that in the case where the connection matrix is suitably engineered to possess real eigenvalues $\varepsilon$ with orthogonal eigenvectors, the populations converge to the same value $N_n(t) = -\varepsilon$ and all oscillators tend to synchronize their phases. This case is numerically illustrated in Supplementary Fig. 9e,f.

In the anapole chain, coupling coefficients can be altered by acting on the geometric topology of the array. A thorough analysis on this topic goes beyond the scope of this article, and is deferred to a future specialized work. We will here consider a simple and illustrative case, where we shift by 6° the anapole array along the direction $y$ (see Fig. 6a). In this configuration the emission dynamics exhibits the appearance of a larger number of synchronized frequencies in the spectrum (see Fig. 6e), and the stable emission of a pulse train of period 1.65 ps (see Fig. 6c) and pulse duration 145 fs (see Fig. 6d). Figure 6b shows a section along the $z$ plane of the energy energy distribution in the structure, and well illustrates the mechanism of formation of the energy pulses inside the waveguide channel by the complex interactions among the different anapoles.

By comparing the emission spectra of Figs 5g and 6e we observe that the dynamics of mode locking in the anapole nanolasers involve the selective locking of different spectral frequencies that vary in number and position according to the spatial configuration considered (linear in Fig. 5 versus shifted array in Fig. 6). This result opens interesting possibilities to use complex anapole arrays as sources for future optical integrated neurocomputers[55].

The mode-locking dynamics of Figs 5 and 6 is a characteristic feature of anapoles that is not observed in conventional nanoparticle lasers. Supplementary Fig. 10 reports the results for nanoparticles with diameter 448 nm, arranged in the same geometry of Fig. 5. In this case there is no pulse generation: the

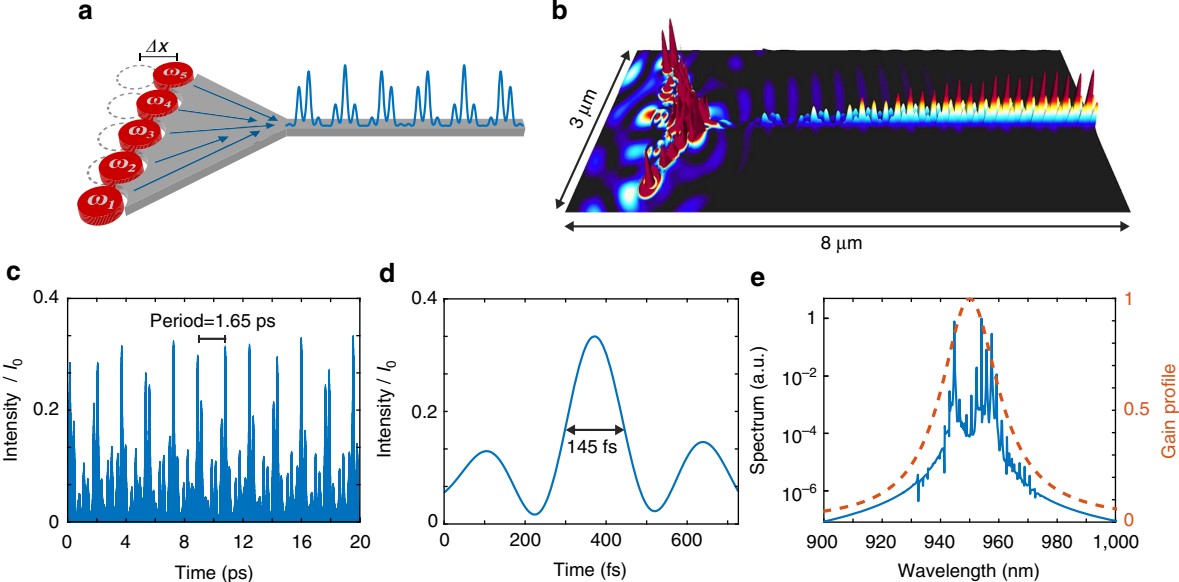

**Figure 6 | Ultrafast pulse engineering by tailoring the anapole geometry.** (**a,b**) Modified anapole array configuration where the nanoparticles from the symmetric structure in Fig. 5 have been shifted of 6° along y. (**b**) Shows a section of the electromagnetic energy steady-state distribution. (**c,e**) Steady-state electromagnetic intensity inside the waveguide, characterized by the formation of 145 fs long pulses repeating within a 1.65 ps period. The pulse profile, as shown in **d**, is characteristic of a system with a large number of synchronized frequencies. The intensity in **a** is normalized to the intensity of the electromagnetic field inside the nanodisk.

collective emission does not mutually synchronize and very little energy is coupled into the nanowire.

## Methods

**Scattering calculations.** All scattering efficiencies for 3D structures have been computed by means of the freely available package DDSCAT[33]. As an input condition, we considered a monochromatic plane wave illumination directed along the nanodisk axis and with polarization perpendicular to the nanodisk axis (see Supplementary Fig. 7).

**MB-FDTD simulations.** We used a fully vectorial, 3D implementation of the MB equation, where the gain material is described by a quantum-mechanical four-level system[56–58]. The interaction between the FDTD electromagnetic fields $\mathbf{E}$, $\mathbf{H}$ and the atoms composing the amplifier is expressed by the polarization response of the material $\mathbf{P} = \mathbf{P}_{\text{lin}} + \mathbf{P}_{\text{Nlin}}$ that is decomposed into a linear $\mathbf{P}_{\text{lin}}$ and nonlinear contribution $\mathbf{P}_{\text{Nlin}}$. The latter is related to the atomic transitions in the amplifier by the following set of MB equations:

$$
\begin{cases}
\partial_t \mathbf{H} = -\frac{1}{\mu_0} \nabla \times \mathbf{E}, \\
\partial_t \mathbf{E} = \frac{1}{\varepsilon_0}[\nabla \times \mathbf{H} - \partial_t(\mathbf{P}_{\text{lin}} + \mathbf{P}_{\text{Nlin}})], \\
\mathbf{P}_{\text{lin}}(t) = \varepsilon_0 \int dt' \chi(t-t') \mathbf{E}(t')
\end{cases}
\quad
\begin{cases}
\mathbf{P}_{\text{Nlin}} = eq_0 N_a [S_1 \hat{\mathbf{x}} + S_4 \hat{\mathbf{y}} + S_9 \hat{\mathbf{z}}], \\
\partial_t S_l = \sum \Gamma_{lm} S_m - \frac{1}{\tau_l}\left[S_l - S_l^{(0)}\right], \\
\Gamma_{lm} = \frac{j}{2\hbar} \text{Tr}\{\mathcal{H}[\lambda_l, \lambda_m]\}
\end{cases}
\tag{3}
$$

where $q_0$ is the quantum displacement length, $e$ is the electric charge, $N_a$ represents the density of polarizable atoms in the excited state and $\tau_j$ are the atomic decaying constant rates. In equation (3), $S_j$ is the $j$th component of the Bloch coherence vector $\mathbf{S}$ that groups different complex-valued elements of the density matrix $\rho_{mn}$ into a real-valued vector state. A detailed derivation of equation (3), including the expressions for the coupling $\Gamma_{lm}$ and Hamiltonian $\mathcal{H}$, components can be found in refs 57,58. As discussed in ref. 41, our MB implementation explicitly considers a realistic source of quantum noise, while the density of excited atoms $N_a$ plays the role of an external pumping rate. The optical properties of InGaAs were taken from published experimental data[32,59]. The dephasing time for the active levels of InGaAs is set as $\tau = 40$ fs, following experimental measurements available in the literature[18,34]. Material dispersion was included in the FDTD algorithm by employing a representation based on a multipolar expansion that correctly represents the media response at visible wavelengths[58].

**Data Availability.** The data that support the findings of this study are available from the corresponding author upon reasonable request.

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

## Acknowledgements

A.F. acknowledges funding support from KAUST (Award No. OSR-2016-CRG5-2995). For the computer time, we have used the resources of the KAUST Supercomputing Laboratory and the Redragon cluster of the PRIMALIGHT group.

## Author contributions

J.S.T.G. and A.E.M. carried out scattering calculations. J.S.T.G. performed first-principle parallel simulations. Y.S.K. and A.F. supervised the research. All authors contributed to the analysis of data. All authors contributed equally to manuscript preparation.

## Additional information

**Competing interests:** The authors declare no competing financial interests.

