## [Peer Review File · Nature Communications]

Reviewers' comments:

Reviewer #1 (Remarks to the Author):

Paper: Juan S. Gongora, Andrey E. Miroshnichenko, Yuri S. Kivshar and A. Fratalocchi, "Anapole nanolasers for mode-locking and ultrafast pulse generation", Nature Communications (2016).

This paper reports a novel type of laser based on a scattering-free anapole (superposition of electric and toroidal dipoles) in a gain medium. The authors claimed that the anapole nanolaser can enable efficient coupling to waveguides and a new mechanism of mode-locking for ultrafast laser pulse generation.

In general, the anapole nanolaser concept is very interesting and can potentially lead to many exciting applications. I think the idea is sufficient to warrant publication if the authors convincingly demonstrate its unique capability compare to other nanolaser systems. However, I am not yet fully convinced that the theory and simulation results have shown that. The following are my specific comments:

1. Based on my understanding of the paper, anapole nanolasers are unique because stimulated emission occurs at a scattering-free state which does not provide any optical feedback. Can the authors clarify how it is conceptually different from other systems such as a laser that relies on the dark mode, or bound state in the continuum (BIC)?

2. From the laser theory perspective, does the anapole nanolaser still require population inversion? Which parameters will dictate the threshold carrier density? And what would be the corresponding cavity quality factor? How does the emission get amplified without any optical feedback?

3. In the optical router (Fig. 4), can the authors provide more details on how the pump beam can selectively excite the anapole with field polarization perpendicular to the pump polarization? While I understand that the pump is chosen at a different frequency corresponding to the maximum scattering, the symmetry of the mode patterns at this frequency and how it couples to the specific spatial field of the anapole should be clarified. The authors showed a four orders of magnitude improved efficiency in selective coupling to the waveguides compare to a InGaAs spherical nanoparticle. However, the coupling efficiency generally depends on the mode matching, so I am not sure if one single comparison (for a specific geometry) is convincing enough to prove the superiority of the anapole. The authors should also comment on its advantage compare to other similar systems such as a metallo-dielectric nanolaser-waveguide design or a plasmon laser circuit.

4. For the mode-locking part, the authors stated that the anapole chain performs like weakly coupled nonlinear oscillators, and the weak coupling is due to the spatial overlap of the anapole emitted fields. While I agree that the system has the advantage of not needing external design elements, I wonder how it is different from other integrated phase-locked lasers that rely on evanescent coupling of multiple laser sources?

5. For practical application, electrically pumped nanolasers are preferred. Can the authors comment on any possible route towards electrical injection without perturbing the anapole? Similarly, will the anapole mode be severely affected by having a realistic lattice-matched substrate beneath it?

-----END-----

Reviewer #2 (Remarks to the Author):

The authors present a numerical study of "anapole lasers" and suggest a number of applications, including couplers, splitters, and mode-locking. Since anapoles correspond to non-radiating configurations, such systems are uncoupled to free-space allowing amplification of the near-field. The paper deals with a timely topic, which is currently attracting substantial attention in the nanophotonics research community. The manuscript introduces intriguing ideas, however, the set of results presented is not sufficient to assess the importance of the work. In particular:

1. The main issue is how good an approximation to an ideal anapole is the configuration of Figs 3c-e. The authors should provide a multipole decomposition of this configuration including electric quadrupole, magnetic dipole and quadrupole in addition to the electric and toroidal dipole contributions.
2. In Fig. 2c, it is not clear under which conditions the scattering cross-section is calculated. Is this assuming plane wave illumination?
3. In Fig. S2, the authors present a multipole decomposition for the ideal lossless configuration. They should include higher order multipoles (see comment 1) and also present results for the case of realistic losses in the system.
4. In Fig. S3, it is shown that the scattering cross-section does not depend strongly on the presence of losses. The authors should provide a justification for this. Also, why does the scattering cross-section retain high values even at the anapole resonance?
5. The authors suggest that changing the geometry of the anapole chain from the waveguide allows to control the temporal dynamics of the system. The authors should discuss the main mechanism for this behaviour. For example, is it due to a change in the interactions within the chain, or simply an interference effect of the emitted fields (or both)?
6. In page 10, the authors mention a statistical campaign of simulations. It would be useful to provide some more information on this.
7. Prior art is not properly referenced. For example, active sources of electromagnetic fields uncoupled or weakly coupled to free-space have been suggested before, see for example Phys. Rev. Lett. 90, 027402 (2003) and Nat. Photonics 2, 351–354 (2008). Sub-diffraction-limit laser sources have been studied in many works, such as Nature 461, 629 (2009) and Nature Mater. 10, 110 (2011). Moreover, an active source based on toroidal modes was introduced in Sci. Reports 3, 1237 (2013). The dynamic anapole was introduced in J. Phys. A: Math. Gen. 28, 4565–4580 (1995) and first observed in Sci. Rep. 3, 2967 (2013). For a history of the field see Nature Mater. 15, 263 (2016) and references therein.
8. The authors should discuss advantages and disadvantages of the suggested scheme compared to other approaches in the literature (see comment 7).
9. For the cases of Figs. 4-6 the authors should normalize the output power to the input (pump) power.

Overall, this is interesting work that could merit publication in Nature Communications, but at this stage there is simply not enough information to assess its significance.

Reviewer #3 (Remarks to the Author):

This is an interesting paper reporting on nanoscale lasers based on tightly-confined anapoles. The authors show on the basis of semiclassical simulations that it is possible to engineer a nanolaser running on anapole modes. In a design using conventional InGaAs nanodisks they suggest on-chip sources to efficiently couple light into waveguide channels with high intensity and a route to generate ultrafast sub-picosecond pulses.

The paper is well written and merits being considered for publication. However, before publication, the authors are required to sufficiently address the following points:

(1) The laser dynamics seems to be dominated by an anapole mode. What is the radiating (far-field) mode? Which pattern will be coupled out? Is there a quadupolar component?

(2) The periods of pulses in Fig. 5 and Fig. 6 are 190fs (seemingly from simulations) and 1.65ps (experimentally measured). Why? This analysis is very important is very important for an understanding of the process and the authors should provide an explanation to fully justify their experimental observation.

(3) Fig.1b shows a 3 -level diagram, but the authors state that they used a 4-level model in the method. Which is correct? Moreover, further details should be provided about the method beyond (a) simple reference(s).

(4) The authors should discuss and cite a number of recent papers reporting on (the spatio-temporal dynamics of) nano-lasing and amplified spontaneous emission on and with non-radiating ('dark') near-field modes in systems such as in

- Nano-fishnet metamaterials: S. Wuestner et al. Phys. Rev. B 85, 201406 (R) (2012).

- 'Stopped-light lasing' in nano-plasmonic waveguides: T. Pickering et al., Nature Communications 5, 4972 (2014).

Response to Reviewers' comments

and a summary of the changes made in the manuscript

Referee #1:

Paper: Juan S. Gongora, Andrey E. Miroshnichenko, Yuri S. Kivshar and A. Fratilocchi, "Anapole nanolasers for mode-locking and ultrafast pulse generation", Nature Communications (2016). This paper reports a novel type of laser based on a scattering-free anapole (superposition of electric and toroidal dipoles) in a gain medium. The authors claimed that the anapole nanolaser can enable efficient coupling to waveguides and a new mechanism of mode-locking for ultrafast laser pulse generation. In general, the anapole nanolaser concept is very interesting and can potentially lead to many exciting applications. I think the idea is sufficient to warrant publication if the authors convincingly demonstrate its unique capability compare to other nanolaser systems. However, I am not yet fully convinced that the theory and simulation results have shown that. The following are my specific comments:

Based on my understanding of the paper, anapole nanolasers are unique because stimulated emission occurs at a scattering-free state which does not provide any optical feedback. Can the authors clarify how it is conceptually different from other systems such as a laser that relies on the dark mode, or bound state in the continuum (BIC)?

Our reply:

We thank Referee #1 for the positive assessment, as well as for many comments that helped us to significantly improve the outreach of this work. There are many differences between an anapole laser and lasers relying on dark modes or BIC states, as we summarize in the following.

Nanolasers based on dark modes are related to resonance effects, as discussed in the following articles for nonlinear metamaterials and SPASERS:

[A1] Wuestner, Sebastian and Hamm, Joachim M. and Pusch, Andreas and Renn, Fabian and Tsakmakidis, Kosmas L. and Hess, Ortwin, "Control and dynamic competition of bright and dark lasing states in active nanoplasmonic metamaterials", *Phys. Rev. B* 20, 201406 (2016),

[A2] Bergman, D. J. & Stockman, M. I. Surface Plasmon Amplification by Stimulated Emission of Radiation: Quantum Generation of Coherent Surface Plasmons in Nanosystems. *Phys. Rev. Lett.* **90**, 27402 (2003).

In an ideal dark laser, dark states are uncoupled to radiation and non-radiating transitions are necessary to transfer energy from the optical amplifier to the dark resonance, as discussed in the original concept of SPASER introduced in [A2] and in the metamaterial laser theoretically analyzed in [A1]. Non-radiating transitions typically require the presence of plasmonic resonances in (most of the cases noble) metals. These resonances, such as localized dark plasmon polaritons, allow for non-radiative energy coupling from an optical amplifier to a dark plasmon.

Contrary to a dark-mode, the anapole is a non-resonant state of the system, and it is typically associated to a strong minimum in the scattering from the material. The anapole is generated by the destructive interference of two radiating components --an electric dipole and a toroidal dipole-- that exist at the same frequency and that cancel each other in the far field. Unlike a dark-resonance, the anapole state allows for radiative mechanisms of direct excitation, as discussed in Ref. [16] of the revised paper:

[16] Wei, L., Xi, Z., Bhattacharya, N. & Urbach, H. P. Excitation of the radiationless anapole mode. *Optica* **3**, 799 (2016).

In the anapole laser, the energy down-converted by stimulated emission at the anapole frequency directly excites the electric and toroidal dipoles that interfere destructively, generating a radiation-less state that does not coexist with additional radiating components and that is observed in a purely dielectric structure. This is a unique working principle of this laser that is not observed with other types of systems relying on resonance effects and/or non-radiative transitions. To clarify this statement quantitatively, we analyzed in more detail the profile of the energy emitted from the anapole laser (Fig. 3 of the revised text). Figure a1 shows the multipolar decomposition (ED: electric dipole, TD: toroidal dipole, MD: magnetic dipole, EQ: electric quadrupole, MQ: magnetic quadrupole and C_{abs} : absorption cross section) of the electromagnetic field in the InGaAs nanodisc at the anapole frequency for different values of the losses ($k=0$: ideal lossless material, $k=0.15$, real absorption of InGaAs):

Fig. a1: **Effect of losses in the Anapole laser (a)-(e)** Multipole decomposition for an InGaAs cylinder of height $h=100$ nm and radius 220 nm as a function of the incident wavelength λ and of the material losses k . The multipole components are: (a) Electric Dipole (ED), (b) Toroidal

Dipole (TD), (c) Magnetic Dipole (MD), (d) Electric Quadrupole (EQ) and (e) Magnetic Quadrupole (MQ). (f) Absorption cross-section.

Figure a2 shows the multipole decomposition for an ideal anapole in InGaAs in the case of no losses $k=0$.

Fig. a2 **Multipole analysis of an ideal anapole state.** The multipole components are computed in Cartesian coordinates and they correspond to: Electric Dipole (blue line), Toroidal Dipole (orange line), Magnetic Dipole (green line), Electric Quadrupole (cyan line), and Magnetic Quadrupole (violet line). At the anapole frequency (dashed vertical line), toroidal and electric dipole cancel each other producing the radiation-less state.

To quantify the degree of similarity between our laser emission and the ideal anapole state, we computed the cross-correlation coefficients of their spatial distributions. More specifically, we compared the ideal anapole of Fig. a2 and the electromagnetic distribution shown in Fig. 3 of the main text, which was calculated considering a real InGaAs nanodisk laser. The cross-correlation coefficient is 96.1%, which shows that the anapole laser allows to create a practically ideal radiation-less anapole state, without the need of any external radiating component, but just destructive interference of specific multipoles that are suitably excited by stimulated emission. The high cross-correlation value observed in our numerical experiments originates from the fact that, as shown in Fig. a1, the presence of losses diminishes (of almost the same factor) the energy of all multipolar components and increases the absorption scattering cross-section of the nanodisk. This implies that the energy difference that is not coupled to ED and TD dipolar components is absorbed inside the nanodisk, without altering the nature of the state being amplified that is 96.1% close to an ideal anapole.

We revised the main text on lines 52-56, 171-175, and added Refs. [A1]-[A2] as Refs. [29,28]. We also added Figs. a1-a2 as Supplementary Figures 2-3, which are commented in a new section titled “Effects of losses in the Anapole laser” in the Supplementary Information.

The comparison with nanolasers based on BIC states is also timely, as these systems have recently attracted conspicuous interest:

[A3] Hsu, C. W., Zhen, B., Stone, A. D., Joannopoulos, J. D. & Soljačić, M. Bound states in the continuum. *Nature Reviews Materials* **1**, 16048 (2016).

[A4] Kodigala, A. *et al.* Lasing action from photonic bound states in continuum. *Nature* **541**, 196–199 (2017).

A fundamental difference between our system and nanolasers emitting on BIC states lies in the fact that an anapole laser is compact and localized. On the contrary, standard BIC implementations require extended structures, where the formation of a BIC state is either achieved by symmetries in the photonic band structure or by engineering supercavities.

Another important difference consists in the fact that in a BIC laser, localized BIC states usually exist and compete with a large number of propagating modes. In the case of the anapole-based nanolaser, conversely, the non-radiating state dominates the system response and radiating components are negligible, as discussed previously.

We revised the manuscript on lines 56-58, and added Refs. [A3]-[A4] as Refs. [31,30].

Referee #1:

From the laser theory perspective, does the anapole nanolaser still require population inversion? Which parameters will dictate the threshold carrier density? And what would be the corresponding cavity quality factor? How does the emission get amplified without any optical feedback?

Our reply:

Our first principle simulations show that the lasing characteristics of the anapole laser (emission and linewidth vs pumping rate), follow the same behavior of a standard laser. The system requires population inversion, in order for the optical amplifier to provide the necessary gain to the structure, as depicted in revised Fig. 1. The carrier threshold density to lasing, as observed in Fig. 3 of the main text, depends on the pumping rate ρ_0 , which is proportional to the population inversion density, or equivalently, to the input pumping rate. By increasing the latter, we increase ρ_0 and control the laser threshold condition.

As discussed above, the anapole state does not originate from a resonance of the system. This specific condition makes the definition of the quality factor Q for an anapole an ill-posed problem. The quality factor Q , in fact, is defined for resonant states that radiate electromagnetic energy in time proportionally to $e^{\{i\omega t - \frac{\omega_0 t}{2Q}\}}$, being ω_0 the frequency of the electromagnetic resonance. This implies that in the scattering cross section of the system at the resonance ω_0 , an observer would measure a Lorentzian peak with full width half maximum equal to ω_0/Q . An anapole state, conversely, does not possess a Lorentzian shape and is observed in points of the

spectrum where the scattering cross section vanishes. This implies that it is not possible to associate a Q factor to this particular state.

The amplification of anapole states does not require optical feedback. This depends on the fact that the anapole is a localized state and, at such, it can be directly amplified by stimulated emission. The steady state amplification of the anapole results from a balance between field-enhancement originated by the gain material and the losses of the system, including absorption and near-field energy leakage. This is the same scenario that is observed in standard nanolasers such as, e.g., core-shell spasers:

[A5] Noginov, M. A. et al. Demonstration of a spaser-based nanolaser. *Nature* 460, 1110–1112 (2009).

or semiconductors nanolasers, Ref. [18] of the revised manuscript:

[18] Chen, R. *et al.* Nanolasers grown on silicon. *Nat Photon* 5, 170–175 (2011).

We revised the main text on lines 155-158, 130-135, 149-154, and added Ref. [A5] as Ref. [42] in the revised text.

Referee #1:

In the optical router (Fig. 4), can the authors provide more details on how the pump beam can selectively excite the anapole with field polarization perpendicular to the pump polarization? While I understand that the pump is chosen at a different frequency corresponding to the maximum scattering, the symmetry of the mode patterns at this frequency and how it couples to the specific spatial field of the anapole should be clarified.

Our reply:

We apologize with the Referee if the manuscript was not clear on this point. The anapole symmetry selection results only from the polarization of the pump beam. The anapole state, in fact, is originated from the interference of an electric dipole and toroidal dipole: the latter has cylindrical symmetry, while the former has a strongly asymmetric shape elongated in the spatial direction perpendicular to the incident polarization. The result of this interference, as illustrated in Fig. 4 of the main text and in Supplementary Fig. 7, shows a three-lobe asymmetric shape perpendicular to the electric field polarization, with energy localized at opposite spatial sides on the nanodisc. As a consequence to this result, the total symmetry direction of the excitation of the electric dipole is orthogonal to the input polarization, as shown in Supplementary Fig. 4 of the main text. When we nonlinearly amplify the anapole in our laser, the input pump polarization selectively excites the corresponding symmetry direction in the electric dipole, leading to different energy distributions in the final anapole state. In order to clear this point in the text, we revised Fig. 4 of the main text by adding a clear indication of the polarization selection mechanism in Panel a:

Revised Fig. 4

We then revised the main text on lines 186-194 and explained in more detail the polarization selection rule of the anapole state amplified in the laser.

Referee #1

The authors showed a four orders of magnitude improved efficiency in selective coupling to the waveguides compare to a InGaAs spherical nanoparticle. However, the coupling efficiency generally depends on the mode matching, so I am not sure if one single comparison (for a specific geometry) is convincing enough to prove the superiority of the anapole. The authors should also comment on its advantage compare to other similar systems such as a metallo-dielectric nanolaser-waveguide design or a plasmon laser circuit.

Our reply:

We agree with the Referee that another comparison can help to demonstrate the superior performance of this system. The nanosphere in Fig. 4, for the same radius of the cylinder of the anapole, possesses a strong scattering peak at the same frequency of the anapole, and therefore permits a comparison in similar optical conditions.

Following the Referee suggestion, we revised the manuscript and included a direct comparison with another integrated structure, composed by a cylindrical nanolaser emitting at a frequency with strongly dipolar energy emission. The latter well couples with the fundamental waveguide mode of the nanowire, and therefore constitutes an interesting benchmark for the anapole laser. In our analysis, we considered an InGaAs cylinder with the same geometrical parameters of the anapole laser (radius $d=220$ nm and height $h=100$ nm), but emitting at the wavelength 1125 nm, located on a resonant peak as in a conventional nanolaser structure. The nonlinear emission of the semiconductor is set to this value by considering an Indium fractional concentration $x=0.32$.

Changing the In concentration of the semiconductor induces a slight variation of the complex refractive index $n(\lambda)$, which we have explicitly considered in our simulations. Due to the material variation, the scattering spectrum of the InGaAs cylinder is slightly modified, as can be seen in Fig. a4:

Figure a4: Scattering cross-section for a $\text{In}_{0.32}\text{Ga}_{0.68}\text{As}$ cylinder. The In concentration $x=0.32$ corresponds to an emission at the wavelength 1125nm. As a reference, we report the scattering efficiency for the proposed anapole laser (blue dotted line), based on $\text{In}_{0.15}\text{Ga}_{0.75}\text{As}$.

In this configuration, the nanolaser emits on a strongly scattering dipolar mode, which is dominated by an electric dipole spatial distribution. This is further illustrated in Fig. a5, where we report the volumetric plot of the electromagnetic energy at the emission wavelength.

Figure a5: Electromagnetic energy of a cylinder emitting on a dipolar scattering state.

As can be seen from the figure, most of the electromagnetic energy is radiated into space, while a small percentage of energy is coupled into the waveguides via butt-coupling. In addition to that, the system couples energy in all the waveguides, as can be easily observed by Fig. a6:

Figure a6: Surface plot of (x, y) section of the electromagnetic energy in logarithmic color scale.

The energy coupled in the nanowire channel in this configuration is 15 times lower (more than one order of magnitude) than the one observed with the anapole laser, thus proving the superior performance of this system. This result depends on the fact that the anapole laser takes advantage from the mechanism of side waveguide coupling, which is superior to butt coupling with propagating modes. The latter process relies on perfect mode matching, which is very challenging to achieve with integrated nanostructures.

Compared to plasmonics systems, the main advantage of the anapole lies in direct integrability and fully CMOS compatibility for silicon photonics circuits. The anapole laser does not employ noble metals and at such the cost is also reduced for this structure. In addition to these points, another important advantage lies in the fabrication: the anapole has an all-dielectric structure that can be easily controlled by e-beam lithography with high nanoscale precision.

We revised the text on lines 207-213 added Figs. a4-a6 as Supplementary Fig. 8.

Referee #1:

For the mode-locking part, the authors stated that the anapole chain performs like weakly coupled nonlinear oscillators, and the weak coupling is due to the spatial overlap of the anapole emitted fields. While I agree that the system has the advantage of not needing external design elements, I wonder how it is different from other integrated phase-locked lasers that rely on evanescent coupling of multiple laser sources?

Our reply:

In phase-locked laser arrays, each laser typically emits at the same frequency and the phase locking condition affects the spatial distribution of the emitted radiating field. This a powerful technique that is normally used to control the spatial distribution the laser, as discussed, e.g., in in Ref. [51] of the revised text, which we report here for completeness:

[51] Kao, T.-Y., Reno, J. L. & Hu, Q. Phase-locked laser arrays through global antenna mutual coupling. *Nature Photonics* 10, 541-546 (2016).

In the anapole laser array, conversely, each nanolaser emits at a different frequency and the phase locking condition is spontaneously developed in the time dynamics of the field, opening the possibility to control at the ultrafast scale the time evolution of the electromagnetic energy emitted from the laser. This difference makes an anapole laser array more similar to an oscillatory neural network, which has been suggested as the basic unit for neurocomputers, i.e., a future generation of computers that are based brain functions, as introduced in:

[A6] Hoppensteadt, F. C. & Izhikevich, E. M. Oscillatory Neurocomputers with Dynamic Connectivity. *Phys. Rev. Lett.* 82, 2983–2986 (1999).

To provide more details on this scenario, and clearly illustrate the differences with a more conventional laser array, we revised the section related to mode-locking of anapoles (lines 224-228, 235-311), employing a simplified model that explains the main dynamics of the Anapole system, which is obtained by generalizing the theory of multi-modal laser systems, as discussed in e.g., Section II of:

[A7] Javaloyes, J., Mandel, P. & Pieroux, D. Dynamical properties of lasers coupled face to face. *Physical Review E* 67, (2003).

to the multi-frequency emission regime. We added also one new section in the supplementary material titled “Nonlinear dynamics of the anapole chain”, which explains the derivation of the model and a new supplementary figure 10, which contains simulation results on the nonlinear phase-locking dynamics of the anapoles. We also added Refs. [A6,A7] as Refs. [55,53].

Referee #1:

For practical application, electrically pumped nanolasers are preferred. Can the authors comment on any possible route towards electrical injection without perturbing the anapole?

As the Referee correctly points out, electrical pumping presents several advantages, especially with reference to integrated applications. Our choice of InGaAs is primarily driven by the large-scale use of this material in electrically pumped diode lasers, such as:

[A8] Zhou, Z., Yin, B. & Michel, J. On-chip light sources for silicon photonics. *Light Sci Appl* 4, e358 (2015).

To realize an electrically pumped anapole laser, the main point is to design a semiconductor heterostructure that ensures three-dimensional confinement of the anapole mode. As an alternative to a typical vertical cavity setup, a possible solution is to consider a core-shell geometry, such as the one proposed in Ref. [18]. As discussed in:

[A9] Liu, W., Zhang, J., Lei, B., Hu, H. & Miroshnichenko, A. E. Invisible nanowires with interfering electric and toroidal dipoles. *Optics Letters* 40, 2293 (2015),

anapole states can be excited in core-shell structures by acting on the multilayer thicknesses. By employing this strategy, an electrically pumped anapole laser can be composed of InGaAs/GaAs or InGaAs/InP heterostructures. This structure, after adding top and bottom electrodes for electric carriers injection, can provide an initial setup for an electrically pumped anapole laser, where the anapole state is generated in the intrinsic layer InGaAs.

Another important point is the optical footprint of the electrical circuitry. To this extent, dielectric conductors such as, e.g., indium tin oxide (ITO), are preferred to metal contacts. The use of dielectric contacts helps in minimizing optical losses. An intriguing possibility would be the introduction of graphene-based contact layers, such as the ones proposed in:

[A10] Kim, Y.-H. *et al.* Graphene-contact electrically driven microdisk lasers. *Nature Communications* **3**, 1123 (2012),

which is a promising design for a room-temperature electrically pumped anapole laser.

We revised the manuscript on lines 116-129 and added Refs. [A8-A10] as Refs. [35,26,37].

Referee #1:

Similarly, will the anapole mode be severely affected by having a realistic lattice-matched substrate beneath it?

Our reply:

This is an interesting technological question. If the lattice mismatch affects few lattice sites only, we do not expect any perturbation in the optical properties of the system and in the resulting dynamics of the anapole laser. However, to reduce lattice-mismatch non-idealities, the InGaAs layer can be grown on a lattice-matched substrate, and then transferred on a low-refractive index substrate that is compatible with Si technology. At the proposed $x=0.15$ concentration of Indium, the InGaAs is lattice matched to both InP and GaAs, which could be used as ultra-thin epitaxial growth substrates. By introducing a sacrificial layer (such as, e.g., AlAs), the InGaAs/GaAs or InGaAs/InP substrate is easily transferred to a low-refractive substrate. Such technique is widely used in the fabrication of III-V semiconductor nanolasers on silicon platforms, as discussed in [A8] and in the article:

[A11] Kim, J. *et al.* Ultra-thin flexible GaAs photovoltaics in vertical forms printed on metal surfaces without interlayer adhesives. *Appl. Phys. Lett.* **108**, 253101 (2016).

We revised the main text on lines 106-115 and added Ref. [A11] as Ref. [36].

Referee #2:

The authors present a numerical study of "anapole lasers" and suggest a number of applications, including couplers, splitters, and mode-locking. Since anapoles correspond to non-radiating configurations, such systems are uncoupled to free-space allowing amplification of the near-field. The paper deals with a timely topic, which is currently attracting substantial attention in

the nanophotonics research community. The manuscript introduces intriguing ideas, however, the set of results presented is not sufficient to assess the importance of the work. In particular:

The main issue is how good an approximation to an ideal anapole is the configuration of Figs 3c-e. The authors should provide a multipole decomposition of this configuration including electric quadrupole, magnetic dipole and quadrupole in addition to the electric and toroidal dipole contributions.

In Fig. S2, the authors present a multipole decomposition for the ideal lossless configuration. They should include higher order multipoles (see comment 1) and also present results for the case of realistic losses in the system.

In Fig. S3, it is shown that the scattering cross-section does not depend strongly on the presence of losses. The authors should provide a justification for this. Also, why does the scattering cross-section retain high values even at the anapole resonance?

Our reply:

We thank the Referee for the positive appreciation of our work, as well as for providing timely and extremely valuable comments.

Following the Referee suggestion, we performed a complete multipolar expansion of the field in the presence of losses. The figure b1 shows the multipolar decomposition of the field in terms of fundamental and high order dipoles in the ideal case, while Fig. b2a-e provides the multipolar decomposition in the case of losses, with Fig. b2f showing the corresponding absorption cross section of the system.

Fig. b1 **Multipole analysis of an ideal anapole state.** The multipole components are computed in Cartesian coordinates and they correspond to: Electric Dipole (blue line), Toroidal Dipole (orange line), Magnetic Dipole (green line), Electric Quadrupole (cyan line), Magnetic Quadrupole (violet line). At the anapole frequency (dashed vertical line), toroidal and electric dipole cancel each other producing the radiation-less state.

Fig. b2 **Effect of losses in the Anapole laser (a)-(e)** Multipole decomposition for InGaAs cylinder of height $h=100$ nm and radius 220 nm vs. the incident wavelength λ and material losses k . Multipole components are: (a) Electric Dipole, (b) Toroidal Dipole, (c) Magnetic Dipole, (d) Electric Quadrupole (e) Magnetic Quadrupole. (f) Absorption cross-section.

The role of losses and dissipation in the formation of anapole states is a subject of research interest, as discussed in the recent paper:

[B1] Tribelsky, M. I. & Miroshnichenko, A. E. Giant in-particle field concentration and Fano resonances at light scattering by high-refractive-index particles. *Physical Review A* **93**, (2016).

In our system, as we see from Fig. b2, the net effect of losses in the linear regime is to slightly reduce (of almost the same factor) the energy coupled into all the multipolar components of the electromagnetic field. As shown in Fig. b2f, the energy difference that is not coupled to electromagnetic components contributes to increase the absorption scattering cross section of the system at the anapole frequency. In our particular case, this variation is little as the losses of the system are small. Figure. b3 shows FDTD simulations of the system linear response when a plane wave impinges on the nanodisk supporting an anapole state in the (a) lossless and (b) real InGaAs medium with loss $k=0.15$:

Fig. b3 Effect of losses and scattering suppression of an anapole state. a,b Electromagnetic energy distribution for an In_{0.15}Ga_{0.85}As disc of diameter $d = 440$ nm at the anapole wavelength = 498nm. The difference between the (a) lossless and (b) realistic lossy configuration is minimal. The imaginary refractive index is (a) $k = 0$ and (b) $k = 0.15$.

As seen from FDTD results, the linear response of the system is almost the same in both cases.

More interesting is the effect of losses in the anapole laser, where the anapole state is nonlinearly amplified via stimulated emission. To evaluate this effect quantitatively, we calculated by FDTD simulations to what extent the nonlinearly amplified anapole state of Fig. 3 is close to an ideal anapole, as requested by the Referee. Figure b4 shows the scattered electromagnetic energy in the case of an ideal anapole. This calculation is performed by extracting the anapole state from the scattered fields of the InGaAs nanocylinder in the ideal lossless case with $k=0$.

Figure b4: Scattered Field from an Ideal anapole ($k=0$) for an InGaAs nanodisk of diameter $d=440$ nm and height 100nm.

We then computed the cross correlation between the ideal anapole of Fig. b4 and the electromagnetic distribution shown in Fig. 3 of the amplified anapole in the realistic, lossy InGaAs laser with $k=0.15$. The cross correlation coefficient is 96.1%, which shows that the anapole laser practically amplifies an almost ideal anapole state. This results from the fact that the electromagnetic losses, as shown in Fig. b2f, increase the amount of energy trapped inside the nanodisk, and this mechanism provides an additional contribution to local field enhancement in the nonlinear processes of light-matter interaction, which contributes to amplify a radiationless anapole state in the InGaAs material.

The scattering cross section values associated to the anapole state and presented in Supplementary Fig. 2 well compares with results from published experiments, see, e.g., [B2] and:

[B3] Miroshnichenko, A. E. et al. Nonradiating anapole modes in dielectric nanoparticles. Nat Commun 6, 8069 (2015).

We revised the main text on lines 171-175. We also added an entire new section in the supplementary information titled “Effects of losses in the anapole laser”, adding Figs. b1-b4 as Supplementary Figures 2-5. We also added Refs. [B2-B3] as Refs. [25,24].

Referee #2:

In Fig. 2c, it is not clear under which conditions the scattering cross-section is calculated. Is this assuming plane wave illumination?

Our reply:

The Referee is correct: our linear scattering cross-section calculations were performed by considering a monochromatic plane-wave illumination. We revised the main text on lines 317-319.

Referee #2:

The authors suggest that changing the geometry of the anapole chain from the waveguide allows to control the temporal dynamics of the system. The authors should discuss the main mechanism for this behavior. For example, is it due to a change in the interactions within the chain, or simply an interference effect of the emitted fields (or both)?

Our reply:

The Referee is correct; the temporal dynamics of the system is controlled by changing the interactions among the chain. We followed the Referee suggestion and revised the section related to the ultrafast pulse generation by discussing in more details the mechanism of this behavior. We illustrated this dynamics by resorting to a simplified model of light-matter interaction, which

is obtained by generalizing the theory of multi-modal laser systems, as discussed in e.g., Section II of:

[B4] Javaloyes, J., Mandel, P. & Pieroux, D. Dynamical properties of lasers coupled face to face. *Physical Review E* 67, (2003).

to the multi-frequency emission regime. We added one new section in the supplementary material titled “Nonlinear dynamics of the anapole chain”, which explains the derivation of the model and a new supplementary figure 10, which contains simulation results that illustrate in more details the mechanisms of phase-locking of the anapoles for different coupling conditions. The dynamics of the Anapole chain is very similar to an oscillatory neural network, as introduced in:

[B5] Hoppensteadt, F. C. & Izhikevich, E. M. Oscillatory Neurocomputers with Dynamic Connectivity. *Phys. Rev. Lett.* 82, 2983–2986 (1999),

as basic building block for future neurocomputing architectures, i.e., computers that mimic brain functions.

We revised the main text on lines 227-231 and 238-314 and added Refs. [B2-B3] and Refs. [53,55].

Referee #2:

In page 10, the authors mention a statistical campaign of simulations. It would be useful to provide some more information on this.

Our reply:

We followed the Referee suggestion and revised the text on lines 284-288 furnishing more details.

Referee #2:

*7. Prior art is not properly referenced. For example, active sources of electromagnetic fields uncoupled or weakly coupled to free-space have been suggested before, see for example *Phys. Rev. Lett.* 90, 027402 (2003) and *Nat. Photonics* 2, 351–354 (2008). Sub-diffraction-limit laser sources have been studied in many works, such as *Nature* 461, 629 (2009) and *Nature Mater.* 10, 110 (2011). Moreover, an active source based on toroidal modes was introduced in *Sci. Reports* 3, 1237 (2013). The dynamic anapole was introduced in *J. Phys. A: Math. Gen.* 28, 4565–4580 (1995) and first observed in *Sci. Rep.* 3, 2967 (2013). For a history of the field see *Nature Mater.* 15, 263 (2016) and references therein.*

The authors should discuss advantages and disadvantages of the suggested scheme compared to other approaches in the literature (see comment 7).

Our reply:

We thank very much the Referee for the suggested literature, which we cited as Refs. [28,7-13]
We also found the following relevant additional papers:

[B6] Savinov, V., Fedotov, V. A. & Zheludev, N. I. Toroidal dipolar excitation and macroscopic electromagnetic properties of metamaterials. *Phys. Rev. B* 89, 205112 (2014)

[B7] Kaelberer, T., Fedotov, V. A., Papasimakis, N., Tsai, D. P. & Zheludev, N. I. Toroidal Dipolar Response in a Metamaterial. *Science* 330, 1510–1512 (2010),

which we cited as Refs. [14-15] in the revised paper.

Concerning the advantages of the anapole laser, a first point comes from the lack of radiating components, which allows the anapole laser to shape the beam controllably, opening to applications that are not possible with standard systems based on radiating modes. An example is provided in revised Fig. 4, discussing a spontaneously polarized laser with routing functionalities.

Another advantage of this near-field emitting platform lies in the possibility to engineer anapole-anapole interactions and create nanolasers with advanced functionalities, such as in the example presented in this paper for the generation of ultrafast pulse generation on chip, which is not observed in classical nanolasers and represents an attractive feature of this design, opening also interesting connections with optical neurocomputing.

Another advantage of the anapole laser relies in a compact and integrated all-dielectric structure, which is fully compatible with CMOS technology and Silicon Photonics. This leads to important benefits in the fabrication of the samples, which can be easily controlled in 2D with high precision through electron beam lithography and directly integrated on-chip.

We revised the main text on lines 52-58, 72-74, 182-183, 310-314.

Referee # 2:

For the cases of Figs. 4-6 the authors should normalize the output power to the input (pump) power.

Our reply:

We re-plotted the Figures by normalizing the intensity with respect to the intensity of the electromagnetic field inside the nanodisc, which measures the electromagnetic energy stored in the anapole state within the gain material. This choice provides the adimensional quantity that is typically employed in the study of nonlinear light-matter interactions in optical amplifiers following first principle Maxwell-Bloch equations, see, e.g.:

Moloney, J. V. & Newell, A. C. *Nonlinear Optics*. (Westview Press, 2004).

In the applications described by Figs 4-6, this normalization allows to directly measure the main figure of merit of the interaction, represented by the energy transfer between the anapole and the guided modes in the nanowire channels.

Referee #3:

This is an interesting paper reporting on nanoscale lasers based on tightly-confined anapoles. The authors show on the basis of semiclassical simulations that it is possible to engineer a nanolaser running on anapole modes. In a design using conventional InGaAs nanodiscs they suggest on-chip sources to efficiently couple light into waveguide channels with high intensity and a route to generate ultrafast sub-picosecond pulses. The paper is well written and merits being considered for publication. However, before publication, the authors are required to sufficiently address the following points: The laser dynamics seems to be dominated by an anapole mode. What is the radiating (far-field) mode? Which pattern will be coupled out? Is there a quadupolar component?

Our reply:

We Thank the Referee for the positive appreciation of our manuscript and for the valuable comments and remarks.

The anapole state is radiationless and it is generated by the superposition of toroidal (TD) and electric (ED) dipoles, which cancel each-other in the far-field. Electromagnetic components that do not couple to ED or TD radiate to the far field. To quantify the amount of radiative emission, we performed a full multipolar decomposition of the linear anapole for different values of the losses ($k=0$: ideal lossless material, $k=0.15$, real absorption of InGaAs) as shown in Fig. c1 (ED: electric dipole, TD: toroidal dipole, MD: magnetic dipole, EQ: electric quadrupole, MQ: magnetic quadrupole and C_{abs} : absorption cross section) below.

As can be seen from the figure, the contribution of quadrupolar modes is negligible, and it is further decreased by the introduction of material losses (panels d,e).

The spatial profile of the anapole state, as shown in Fig. 3 of the revised manuscript and in Supplementary Fig. 7, shows a typical three-lobes asymmetric profile perpendicular to the electric field polarization, with energy localized at opposite spatial sides on the nanodisk. To quantify the degree of similarity between the laser emission and the ideal anapole state, we computed the cross-correlation coefficients of their spatial distributions. More specifically, we compared the electromagnetic distribution shown in Fig. 3 of the main text to that of an ideal anapole, calculated from Fig. c1 in the case of no losses ($k=0$). The cross-correlation coefficient is 96.1%, which shows that the anapole laser allows to create an almost ideal radiationless anapole state, without the need of any external radiating component, but just destructive interference of specific multipoles that are suitably excited by stimulated emission. This result originates from the fact that, as shown in Fig. c1, the presence of losses decreases (of almost the same factor) the energy of all multipolar components and increases the absorption scattering cross-section of the nanodisk. This implies that the energy difference that is not coupled to ED and TD components, is absorbed inside the nanodisk. This process does not change the nature of the state being nonlinearly amplified, which is 96.1% close to an ideal anapole.

We revised the main text on lines 171-175 and added Fig. c1 as Supplementary Fig. 3. We also added a new section in the supplementary titled “Effects of losses in the anapole laser”, where we discuss in more detail the formation of the radiationless state in the nonlinear regime of light amplification.

Fig. c1: **Effect of losses in the Anapole laser (a)-(e)** Multipole decomposition for an InGaAs cylinder of height $h=100$ nm and radius 220 nm vs. the incident wavelength λ and of the material losses k . The multipole components are: (a) Electric Dipole, (b) Toroidal Dipole, (c) Magnetic Dipole, (d) Electric Quadrupole (e) Magnetic Quadrupole. **(f)** Absorption cross-section.

Referee #3:

The periods of pulses in Fig. 5 and Fig. 6 are 190fs (seemingly from simulations) and 1.65ps (experimentally measured). Why? This analysis is very important is very important for an understanding of the process and the authors should provide an explanation to fully justify their experimental observation.

Our reply:

We apologize if the manuscript was not clear enough on this point. The results refer to different configurations of anapoles, where the temporal dynamics is controlled by changing the interactions among the anapoles in the chain. We followed the Referee suggestion and revised the section related to the ultrafast pulse generation by discussing in more details the mechanism

of this behavior. We illustrated this dynamics by resorting to a simplified model of light-matter interaction, which is obtained by generalizing the theory of multi-modal laser systems, as discussed in e.g., Section II of:

[C1] Javaloyes, J., Mandel, P. & Pieroux, D. Dynamical properties of lasers coupled face to face. *Physical Review E* 67, (2003).

to the multi-frequency emission regime. We added one new section in the supplementary material titled “Nonlinear dynamics of the anapole chain”, which explains the derivation of the model and a new supplementary figure 10,

Supplementary Fig. 10. Nonlinear dynamics in a chain of anapole lasers. Modal amplitudes $A_n(t)$ (left panels) and phase differences $\Delta\varphi_n(t) = \varphi_n(t) - \varphi_0(t)$ (right panels) as obtained by numerical integration of Eqs. S3 in the supplementary text.

which contains simulation results that illustrate in more details the mechanisms of amplitudes and phase-locking of the anapoles for different coupling conditions, in agreement with our first principle simulations. The dynamics of the Anapole chain is very similar to an oscillatory neural network, as introduced in:

[C2] Hoppensteadt, F. C. & Izhikevich, E. M. Oscillatory Neurocomputers with Dynamic Connectivity. *Phys. Rev. Lett.* 82, 2983–2986 (1999),

as basic building block for future neurocomputing architectures, i.e., computers that mimic brain functions.

We revised the main text on lines 227-231 and 238-312 and added Refs. [C1-C2] as Refs. [53,55].

Referee #3:

Fig.1b shows a 3 -level diagram, but the authors state that they used a 4-level model in the method. Which is correct? Moreover, further details should be provided about the method beyond (a) simple reference(s).

Our reply:

The amplification scheme of the Anapole laser works for both 3 and 4 level systems. In our simulations we used a 4 level scheme, as it is a more general formulation of the Maxwell-Bloch equations. We revised the main text on lines 44-47 and Fig. 1 in order to avoid any ambiguity with the text.

Following the Referee suggestion, we also added a paragraph titled “Maxwell-Bloch Finite-Differences Time-Domain simulations” in the Methods section that provides more details on first principle simulations. We revised the text on lines 320-336.

Referee #3:

The authors should discuss and cite a number of recent papers reporting on (the spatio-temporal dynamics of) nano-lasing and amplified spontaneous emission on and with non-radiating ('dark') near-field modes in systems such as in
- *Nano-fishnet metamaterials: S. Wuestner et al. Phys. Rev. B 85, 201406 (R) (2012).*
- *'Stopped-light lasing' in nano-plasmonic waveguides: T. Pickering et al., Nature Communications 5, 4972 (2014).*

Our reply:

We thank the Referee for the important literature. We included the above-mentioned references as Refs. [29,16]. We also included the interesting article:

[C3] Wuestner, Sebastian, Joachim M. Hamm, Andreas Pusch, and Ortwin Hess. “Plasmonic Leaky-Mode Lasing in Active Semiconductor Nanowires.” *Laser & Photonics Reviews* 9, no. 2 (March 1, 2015): 256–62. doi:10.1002/lpor.201400231,

as Ref [17] of the revised paper.

REVIEWERS' COMMENTS:

Reviewer #2 (Remarks to the Author):

The authors have fully addressed all my comments and have substantially improved their paper. I strongly recommend the manuscript for publication in its current form.

Reviewer #3 (Remarks to the Author):

The authors have significantly amended their manuscript, taking on board the comments/suggestions made in the review report. Moreover, the authors provide a detailed discussion of various points raised by the reviewer in their reply. Altogether the manuscript has been significantly improved and I am now happy to suggest to accept it for publication in Nature Communications.

Response to Reviewers' comments

Reviewer #2 (Remarks to the Author):

The authors have fully addressed all my comments and have substantially improved their paper. I strongly recommend the manuscript for publication in its current form.

Authors:

We thank the Referee for the positive appreciation of our revised manuscript and for the useful suggestions which allowed us to significantly improve the completeness and outreach of our work.

Reviewer #3 (Remarks to the Author):

The authors have significantly amended their manuscript, taking on board the comments/suggestions made in the review report. Moreover, the authors provide a detailed discussion of various points raised by the reviewer in their reply. Altogether the manuscript has been significantly improved and I am now happy to suggest to accept it for publication in Nature Communications.

Authors:

We thank the Referee for the positive assessment of our revision and for the meaningful and timely comments to our work.